# Medium- and Long-Term Outcomes of 597 Patients Following Minimally Invasive Multi-Vessel Coronary Off-Pump Bypass Surgery

**DOI:** 10.3390/jcm14051707

**Published:** 2025-03-03

**Authors:** Magdalena I. Rufa, Adrian Ursulescu, Samir Ahad, Ragi Nagib, Marc Albert, Mihnea Ghinescu, Tunjay Shavahatli, Rafael Ayala, Nora Göbel, Ulrich F. W. Franke, Bartosz Rylski

**Affiliations:** 1Department of Cardiovascular Surgery, Robert Bosch Hospital, 70376 Stuttgart, Germany; adrian.ursulescu@rbk.de (A.U.); samir.ahad@rbk.de (S.A.); ragi.nagib@rbk.de (R.N.); marc.albert@rbk.de (M.A.); mihnea.ghinescu@rbk.de (M.G.); tunjay.shavahatli@rbk.de (T.S.); rafael.ayala@rbk.de (R.A.); nora.goebel@rbk.de (N.G.); bartosz.rylski@rbk.de (B.R.); 2Department of Cardiovascular Surgery, Heart Centre Freiburg University, 79189 Freiburg, Germany; ulrich.franke@uniklinik-freiburg.de

**Keywords:** minimally invasive multi-vessel off-pump coronary artery bypass grafting (MICS CABG), off-pump coronary artery bypass grafting (OPCAB), coronary artery bypass grafting (CABG), coronary artery disease (CAD)

## Abstract

**Background:** Minimally invasive multi-vessel off-pump coronary artery bypass grafting (MICS CABG) through left anterior mini-thoracotomy avoids both extracorporeal circulation and sternotomy and is a very elegant, safe, and effective surgical technique, despite its still-limited adoption in the daily toolkit of cardiac surgeons. The goal of this retrospective, single-centre analysis was to evaluate the long-term outcomes of a large patient cohort undergoing MICS CABG. **Methods**: This study identified 597 consecutive MICS CABG patients from August 2008 to November 2020. We obtained follow-up data by phone or mail. Every patient had a left internal thoracic artery bypass graft. The second and possibly third grafts were radial arteries, great saphenous vein segments, or right internal thoracic arteries. **Results:** The median age was 69 years, and 92.1% were male. The median EuroSCORE II was 1.5. There were eight conversions to sternotomy and none to cardiopulmonary bypass. The total arterial revascularisation was 92.5%, with 90.3% complete. The 30-day mortality was 0.5%. A total of 575 patients (95.8%) were tracked for 8 years on average. A Cox regression analysis found that a left ventricular ejection fraction < 50%, peripheral vascular disease, chronic kidney disease, and a history of cerebrovascular accident independently predicted severe adverse cardiac and cerebrovascular events and late death. The actuarial survival rates for one, three, five, eight, and ten years were 99%, 95%, 91%, 85%, and 80%, respectively. **Conclusions:** In our study group, the technique of MICS CABG has been proven to be a safe and effective surgical revascularisation method, with a low rate of early complications and favourable long-term outcomes in eligible patients.

## 1. Introduction

The target population for myocardial revascularisation has undergone a significant change in recent years. Patients are progressively presenting with more complex coronary artery disease (CAD), as well as multiple comorbidities and frailty. As a result of this change, there is a growing demand for minimally invasive surgical procedures. Minimally invasive multi-vessel off-pump coronary artery bypass grafting (MICS CABG) allows for complete surgical or hybrid myocardial revascularisation without the need for sternotomy or cardiopulmonary bypass (CPB). The criteria for this procedure are comparable to those of conventional coronary artery bypass grafting (CABG) surgery performed through a median sternotomy. Particular attention is paid to the morphology of the coronary arteries and the localisation of the stenoses while performing this approach. MICS CABG, although demonstrating excellent results [1,2,3,4,5], is not widely adopted, and the long-term outcomes of large patient groups are few and inconsistent [6,7,8]. The reported minimally invasive CABG procedures encompass a range of techniques, including off-pump, beating heart on-pump, arrested heart with femoral vessel cannulation [9], no-touch aorta, with proximal anastomoses on the ascending or descending aorta, left lateral anterior thoracotomy under direct vision or robotic-assisted, and total robotic/endoscopic coronary artery bypass [9].

Our centre has 16 years of expertise with MICS CABG, which is an off-pump, no-touch aorta approach with a high rate of complete arterial revascularisation. Selecting patients carefully is crucial to the success of minimally invasive coronary revascularisation. The following conditions currently preclude the use of this technique at our institution: morbid obesity, diffuse and intramyocardial vessels, severe left ventricular dysfunction, emergency situations, severe left pleural fibrosis and adhesions as a result of trauma, surgery, or radiation, and advanced pulmonary disease not able to withstand single lung ventilation.

The objective of this study was to evaluate the clinical outcomes and overall survival of a significant number of patients who underwent minimally invasive multi-vessel off-pump coronary artery bypass grafting procedures at our institution over a 13-year period and also to explore the potential predictors of adverse events.

## 2. Materials and Methods

### 2.1. Study Design

A total of 1315 patients underwent isolated minimally invasive off-pump CABG surgery at our institution between August 2008 and November 2020. We successfully identified all the patients who underwent MICS CABG. An interdisciplinary heart team decided on the treatment plan for every patient according to the specific coronary architecture, degree of complexity of the CAD, and patient’s state of health. Ultimately, the research enrolled 597 patients. The ethics review board approved this study, which was conducted in accordance with the Declaration of Helsinki (as revised in 2013). All the patients provided informed approval.

The following data were incorporated for analysis: demographic characteristics, preoperative risk factors, comorbidities, urgency of surgery, and extent of the CAD. A glomerular filtration rate of less than 50 mL/min/1.73 m^2^ was used to define chronic kidney disease (CKD). In accordance with the urgency of the procedure, the patients were categorised as either elective or urgent (surgery should be performed during the same hospitalisation). A prior myocardial infarction was categorised into four groups: less than 48 h, more than 48 h but less than 21 days, between 21 and 91 days, and over 91 days. Patients were categorised into three groups based on their preoperative left ventricular ejection fraction (LVEF): <30%, 31% to 50%, and >50%. The European System for Cardiac Operative Risk Evaluation (EuroSCORE) II was computed for each patient. The surgery data covered the following topics: the number and nature of the grafts; revascularised heart areas; conversion to cardio-pulmonary bypass and/or sternotomy; rate of transfusion and number of transfused patients; rate of completeness of revascularisation; and whether certain procedures were hybrid planned. Significant events, including atrial fibrillation, new-onset renal insufficiency necessitating dialysis, other organ dysfunctions, cerebrovascular accident (CVA), perioperative myocardial infarction, bleeding, revascularisation through percutaneous coronary intervention (PCI) or bypass revision, duration of intensive care stay and hospitalisation, and 30-day mortality, were included in the postoperative inpatient data collections.

The primary endpoints were the survival rate in the mid- and long-term follow-up, as well as the rate of major adverse cardiac and cerebrovascular events (MACCEs) during follow-up. Recurrent revascularisation via PCI or CABG, CVA, all-cause mortality, and myocardial infarction were all incorporated into the MACCE definition.

When the individual could not be contacted, follow-up information was obtained via letter or phone interviews with their immediate family or with their referring general practitioners. The follow-up rate was 95.8%.

### 2.2. Surgical Technique

Until the end of 2018,we conducted the majority of these surgeries using the left internal thoracic artery (LITA) and a radial artery (RA), preferably from the non-dominant hand, after ensuring that the Allen test was not pathological. In order to guarantee the integrity of their radial arteries prior to surgery, patients underwent routine ultrasound control. In the event that the RA graft was problematic during surgery or that none of the radial arteries could be harvested, a segment of the great saphenous vein (GSV) was harvested. The Vasoview 6 Pro or Vasoview Hemopro 2 endoscopic vessel-harvesting devices (Maquet Inc., Rastatt, Germany) were employed to harvest the RA and/or GSV conduits endoscopically (Figure 1 and Figure 2).

Following the acquisition of a surgical robot, the da Vinci Xi (Intuitive Surgical Inc., Sunnyvale, CA, USA), in 2019, we transitioned to robotic-assisted graft harvesting. The robotic platform provides exceptional visualisation of the internal thoracic arteries, with minimal risk of vessel injury [10]. Additionally, it is significantly less traumatic for the patient than harvesting with long-shaft instruments, as the internal thoracic artery is not exposed with the use of a special retractor. Consequently, a longer ITA graft can be harvested [10]. In light of this, as of July 2019, we began routinely performing MICS CABG using bilateral internal thoracic arteries (BITAs) (Figure 3).

After the necessary bypass graft material is harvested, the arterial grafts are prepared for bypass surgery by internally applying papaverine solution and performing systemic heparinisation. The anastomoses are conducted using an off-pump aortic no-touch technique, which is accompanied by use of vacuum stabilisation devices to ensure stability throughout the anastomotic procedure. Coronary shunts are implemented frequently to mitigate coronary backflow and prevent myocardial ischaemia [10]. The coronary artery anastomosis is made more visible with the use of a blower mister (Figure 4). The LITA is used to revascularise both the left anterior descending artery (LAD) and the diagonal branch. A T-graft construct is employed to connect the bypass grafts from the second graft material, radial artery, saphenous vein, or right internal thoracic artery (RITA) to the LITA in order to provide blood flow to the lateral and/or posterior myocardial regions. VeriQ Doppler flow probes (Medistim, Oslo, Norway) are employed to measure and document the graft flows. In order to alleviate postoperative pain, an intercostal nerve blockade catheter is inserted. Patients are typically extubated prior to departing the operating room and receive 500 mg of aspirin postoperatively.

### 2.3. Statistical Analysis

Categorical data are shown as absolutes and percentages, while continuous variables without a normal distribution are represented as the median and interquartile range (IQR). All the tests were two-tailored, with statistically significant differences defined as a *p* value less than 0.05. A Cox regression model was employed to determine the factors that predict both death and MACCE. We applied the Kaplan–Meier curve to evaluate patients’ survival rates. Our statistical analysis was conducted using SPSS version 28.0 for Windows (SPSS Inc., Chicago, IL, USA).

## 3. Results

Table 1 provides an overview of the demographic profile and clinical characteristics of the research cohort. The study population had a median age of 69 years, with an IQR of 61–76 years. There was a significant male predominance, with 92.1% of the population being male. The average EuroSCORE II was 1.5, with a range of 1 to 2.2. Out of the total sample, 9% or 54 individuals had a pre-existing diagnosis of peripheral vascular disease (PVD). A total of 74 patients (12.4%) had been confirmed with CKD, while 26 patients (4.4%) had experienced a CVA. Out of the total number of patients, 167 (28%) had already undergone a PCI before their surgical myocardial revascularisation. Moreover, 46.1% of the total cases, which amounted to 275, were classified as urgent.

Two primary surgeons performed the procedures. Out of the total of 597 patients, 523 individuals underwent MICS CABG with two grafts, while the remaining 74 patients received three grafts. A LITA bypass was given to all the patients, and in 17.8% of cases, it was the only bypass graft material that was used. As the second graft material, a RA graft, was used in 68.8% of the instances, or 411 cases. Thirty-one patients underwent transplant harvesting with the use of robotic technology. The percentage of total arterial revascularisation was exceptionally high, reaching 92.5%. Cardiopulmonary bypass was not used in any of the eight sternotomy conversions. Lung adhesions prevented the LITA preparation in three instances, intramyocardial course of the LAD in two cases, haemodynamic instability with rising vasopressors doses in two cases, and in one case, technical problems with the anastomoses were the grounds for conversion to sternotomy.

The median duration of ventilation was 2 h, with a range of 0–5 h. The majority of patients were extubated prior to leaving the operating theatre. The rates of transfusion were exceedingly low. A total of 539 patients, accounting for 90.3%, achieved full revascularisation. The operational characteristics are succinctly outlined in Table 2.

Thirteen patients (2.2%) required re-exploration due to bleeding. Two patients (0.3%) had a postoperative CVA, seven (1.2%) had a postoperative myocardial infarction, thirteen (2.2%) had new-onset atrial fibrillation, and six (1.0%) had new-onset renal failure that necessitated dialysis. Three patients (0.5%) died within the first 30 days. One of the patients experienced a sudden asystole on the eighth postoperative day, and cardiopulmonary resuscitation (CPR) treatments failed to save his life. One patient developed hemi colon ischaemia, underwent abdominal surgery, and died from sepsis 10 days later. The third patient experienced new-onset persistent atrial fibrillation. An electric cardioversion could not be performed because a thrombus had already formed in the left atrial appendage. On the sixth postoperative day, the patient spontaneously entered sinus rhythm and, a few seconds later, displayed asystole. CPR was performed promptly, but it did not result in the return of spontaneous circulation, and the patient died. Table 3 summarises the perioperative findings.

The average follow-up period was 7.8 ± 3.5 years, with a median of 8 years and an interquartile range of 5–11 years. A total of 575 patients were followed up on, resulting in a 95.8% completion rate. A total of 22 patients were no longer accessible for follow-up. During the follow-up period, 78 patients, or 13.5%, developed recurrent angina pectoris. Moreover, 189 serious adverse cardiac and cerebrovascular events occurred over the follow-up period, including 25 myocardial infarctions (4.3%), 87 PCI repeat revascularisations (15.1%), 25 CVAs (4.3%), and 100 fatalities, 16 (2.7%) of which were cardiac. Two patients (0.3%) required redo-bypass surgery. The survival rates at one, three, five, eight, and ten years were 99%, 95%, 91%, 85%, and 80%. Figure 5 illustrates the survival rates using Kaplan–Meier estimates.

The Kaplan–Meier analysis revealed an MACCE-free survival rate of 99%, 95%, 91%, 81%, and 72% at 1, 3, 5, 8, and 10 years (Figure 6).

A Cox regression analysis was conducted, revealing that an LV EF below 50%, PVD, CKD, and a history of CVA are independent risk factors for late MACCEs and mortality. Table 4 shows that a previous PCI treatment and the existence of carotid stenosis were additional risk factors for late mortality, but they did not have a meaningful impact on the long-term MACCEs.

## 4. Discussion

The literature reports a variety of techniques for minimally invasive CABG procedures. Many studies addressing this issue involve heterogeneous groups, as multiple techniques are employed across various centres. This retrospective, single-centre, observational study presents the long-term outcomes of a sizable, homogeneous population undergoing MICS CABG, reflecting 13 years of clinical experience. The implied surgical method is an aortic no-touch off-pump coronary artery bypass grafting technique performed via a left anterior thoracotomy. Up to 92.5% of the patients underwent total arterial revascularisation. A very low incidence of in-hospital adverse events and favourable long-term outcomes were observed in this study. Additionally, a history of CVA, a positive history of PVD, the presence of CKD, and an LV EF below 50% were identified as independent risk factors for late MACCEs and mortality.

Our research group has thoroughly examined several facets of our 16 years of experience with minimally invasive direct coronary artery bypass (MIDCAB) and MICS CABG. We have shown that incomplete revascularisation in minimally invasive coronary surgery leads to reduced long-term survival, but this finding lacks statistical significance after risk adjustment [11]. Subsequently, we have analysed and demonstrated that when minimally invasive coronary surgery is incorporated into a hybrid coronary revascularisation (HCR) strategy, the fulfilment of the intended treatment plan, specifically the execution of the PCI step, is essential for achieving complete revascularisation and thereby ensuring an improved survival rate for patients [12]. Finally, we have shown that MICS CABG is as safe and effective as MIDCAB [10].

In our experience, the graft design of MICS CABG changed with the purchase of a medical robot, and thereafter, robotic-assisted harvesting of BITA, thus shifting from RA or SV to RITA and also making the surgery even less traumatic for the patient. We also feel we gained in operating time, although we have not investigated this aspect explicitly, as in many patients with RA harvesting, we harvested the left RA, then we placed the left arm parallel to the torso, and only then could we start with the LITA preparation, and thus the graft harvesting could not be conducted in parallel, but one after the other, therefore prolonging the operating time, whereas with the robotic assistance, once the trocars are placed on the left side, the harvesting of both ITAs can be conducted without any more changes, and after the first learning cases, we have reduced the harvesting time to about 90 min for both ITAs. The method presents a low stroke risk by implying no CPB and by not manipulating the aorta, and it also bears a very low risk of surgical site infections when both ITAs are used in patients who otherwise would present in sternotomy CABG a clear risk of deep sternal wound problems.

Although this study was designed to include all the consecutive patients who underwent MICS CABG at our institution, it still has the limitations associated with a retrospective, single-centre, observational, descriptive analysis conducted over a long period of time. The lack of a control group precludes direct comparison with other CABG techniques and therefore, although this was not the aim of this study, any conclusion regarding the efficacy of the procedure should be drawn very carefully. Bias in regard to patient selection is likely. Routine follow-up assessments for graft patency, including computed tomography or coronary angiography, are lacking, as they are not routinely conducted in our everyday practice.

Even though our results might not be able to be generalised, we think that this information is useful for any centre starting a MICS CABG programme.

In a recent review of minimally invasive coronary artery bypass grafting for multi-vessel coronary artery disease, including a total of 7556 patients from 26 studies, the early mortality and stroke rates were 0.6% and 0.4%, respectively [5]. In our study group, the early death rate was 0.5%, whereas the postoperative CVA incidence was 0.3%. Several studies reported the mid-term outcome [1,3,13,14] and only a few reported the long-term outcomes [6,7].

In the study by Guida et al. on 2500 MICS CABG patients, the authors reported long-term survival rates of 98.8%, 93.6% and 69.1% at 1, 5 and 10 years, respectively [6]. The rates of in-hospital complications were very similar to ours. The percentage of total arterial revascularisation was 14.7% and 8.3% received a MIDCAB procedure. The 30-day mortality was 1% [6].

Sakaguchi et al. presented in their MICS CABG case series a total arterial revascularisation rate of 47.3%, while the aortic no-touch rate was 57% [13]. However, 34.4% received single LITA-LAD bypass. The freedom from MACCEs at 5 years was 89.7% [13].

Guo et al. reported on their 17-year experience with minimally invasive coronary artery bypass grafting on 566 patients, half of whom underwent MIDCAB and the other half MICS CABG [7]. The mean long-term follow-up time was 7 ± 4.4 years. At 12 years, the cohort’s survival rate was 82.2% ± 2.6% [7]. The MACCE-free survival rate was 75.5% ± 3.0% [7]. Our research population had a 10-year survival rate of 80% and a MACCE-free survival rate of 72%, which were somewhat lower, most likely due to the fact that our cohort only included MICS CABG patients and no MIDCAB cases. PVD and left ventricular dysfunction were also identified as risk factors for late mortality and MACCEs [7]. After 12 years, the cumulative frequency of recurrent revascularisation was 14.8% ± 2.5% [7]. During the follow-up period, our research group experienced a total rate of 15.4% for repeat revascularisation.

Slightly over 10% of the procedures were hybrid planned; however, as this study extends over 13 years, commencing in 2008, it remains uncertain whether patients who reported receiving a PCI procedure followed by stent implantation during follow-up were undergoing the originally intended procedure as part of the hybrid treatment or if it resulted from a failed prior stent, unsuccessful bypass, or de novo stenosis. Consequently, we opted to classify the case as incomplete and to categorise the procedure during follow-up as renewed revascularisation, also noting it as MACCE-positive. We have gained significant insights from analysing these data, and we are now thorough in organising and conducting the PCI step in a hybrid case.

The elevated MACCE percentage observed during follow-up in our cohort is primarily due to the significant proportion of PCI and the rate of all-cause mortality. The percentage of myocardial infarction was relatively low at 4.3%, in contrast to the 13.5% of patients reporting recurrent angina pectoris. Unfortunately, we lack detailed data on the PCI procedures performed, including whether they were necessitated by a failed primary stent, failed bypass, or new stenosis resulting from the natural progression of chronic coronary artery disease.

Our study group had a very high rate of total arterial revascularisation, namely 92.5%. In the study group by Guo et al. [7], only 36.4% of the patients received multiple arterial grafts.

Our mid- and long-term results are comparable to those of previously published extensive case reports and prospective randomised studies including patients undergoing traditional sternotomy for on-pump or off-pump CABG [15,16,17]. The CORONARY study indicated a 5-year mortality rate of 13.5% in the on-pump group and 14.6% in the off-pump CABG cohort [15]. In the SYNTAX trial, in the CABG cohort, the 5-year mortality rate was 11.4% [16], and at 10 years, it reached 24% [17]. The 10-year results of the ART trial found a 20.3% late mortality rate in the BITA group and 21.2% in the group where only LITA was used [18]. Chikwe et al. reported 10-year death rates of 33.4% for off-pump surgery and 29.6% for on-pump surgery, respectively [19].

We also compared our data to those reported by groups that are known to have been practicing minimally invasive revascularisation techniques for a long period of time. Repossini et al. documented survival rates of 99.2% at 30 days, 87.1% at 5 years, 84.3% at 10 years, and 79.8% at 15 years in a single-centre study with 1060 patients [20]. Bonatti et al.’s evaluation of minimally invasive cardiac surgery during the past 25 years indicated an in-hospital survival rate of 99% and a 5-year survival rate of 91% [21]. Manuel et al. presented the 20-year results for 271 MIDCAB patients, with survival rates of 91.9%, 84.7%, 71.3%, and 56.5% after 5, 10, 15, and 20 years, respectively [22].

Therefore, our findings underline MICS CABG’s promise as a safe and efficient substitute for conventional CABG surgery.

A recent review study by the University of Ottawa’s workgroup determined, following a comprehensive literature analysis, that MICS CABG is consistently a durable, safe, and practical alternative to sternotomy CABG. The authors asserted that, given the advantageous early outcomes, similar long-term results and graft patency, along with the practicality of its use, MICS CABG represents a significant advancement in cardiac surgery [8].

The increasing number of percutaneous techniques and the decrease in CABG case volumes suggest that MICS CABG may enhance the surgical revascularisation options available, allowing for individualised patient care [5,10]. But because of the technical difficulties and learning curve, MICS CABG is only carried out in a few skilled centres, even with the positive outcomes [5].

Now that drug-eluting stents are getting better all the time, we think that more patients might benefit from an HCR approach that combines MICS CABG with BITA grafting to cover the left coronary system [10]. This approach offers the long-term survival benefits of BITA grafting while mitigating the risk of sternal wound infection, alongside the recognised advantages of employing minimally invasive techniques [8].

Organising a prospective multi-centre international study in centres with significant experience of off-pump and minimally invasive procedures is essential. Obtaining reliable data from this study would provide strong evidence for the establishment and expansion of a minimally invasive CABG programme in more medical centres, given the increasing patient demand [10].

## 5. Conclusions

In our 13 years of experience involving 597 patients, the technique of minimally invasive multi-vessel off-pump coronary artery bypass grafting has been proven to be a safe and effective surgical revascularisation method. It has shown a low rate of early complications and has yielded favourable long-term outcomes in eligible patients.

## Figures and Tables

**Figure 1 jcm-14-01707-f001:**
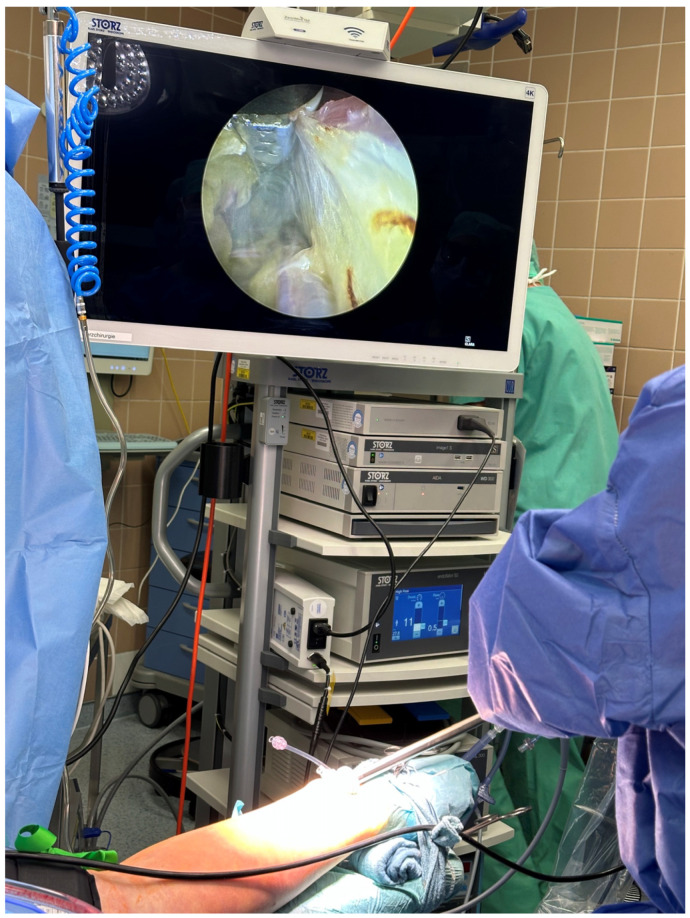
Minimally invasive endoscopic harvesting of the left radial artery.

**Figure 2 jcm-14-01707-f002:**
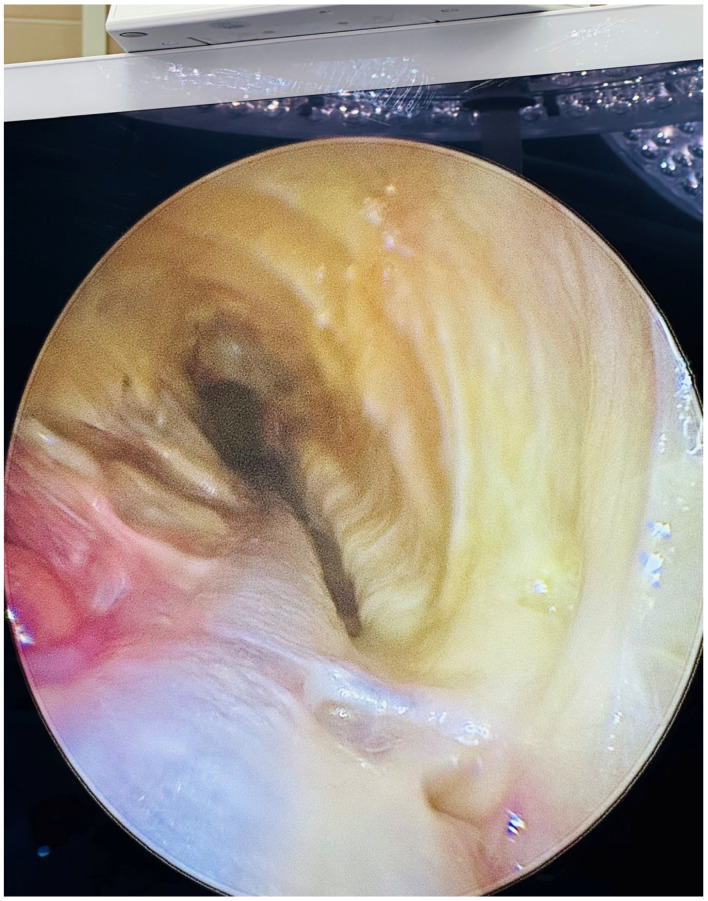
Exposition of the radial artery following fascia preparation.

**Figure 3 jcm-14-01707-f003:**
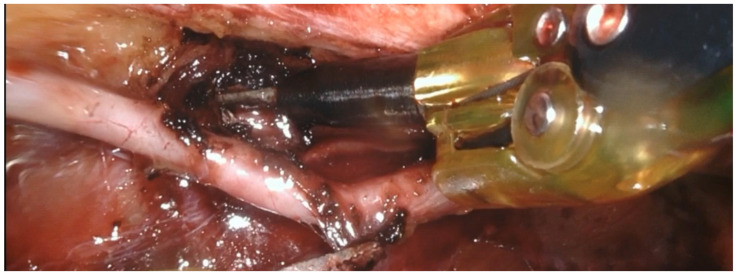
Robotic-assisted harvesting of the internal thoracic artery.

**Figure 4 jcm-14-01707-f004:**
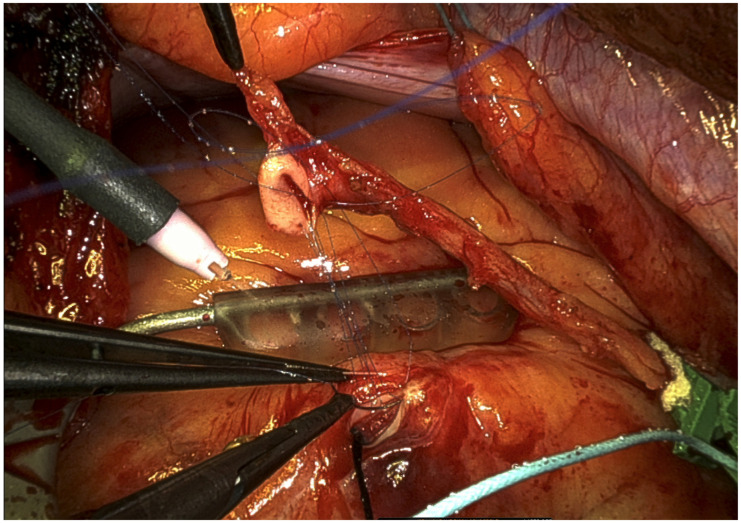
MICS CABG setting during LITA-LAD anastomosis (vacuum stabilisation in place, occlusion clamp on LITA, intravascular coronary shunt in LAD, blower mister).

**Figure 5 jcm-14-01707-f005:**
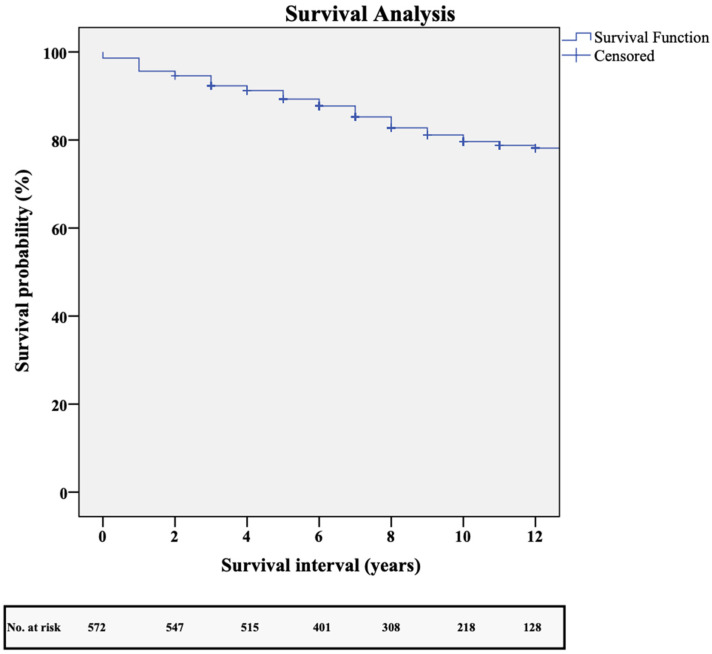
Kaplan–Meier survival analysis.

**Figure 6 jcm-14-01707-f006:**
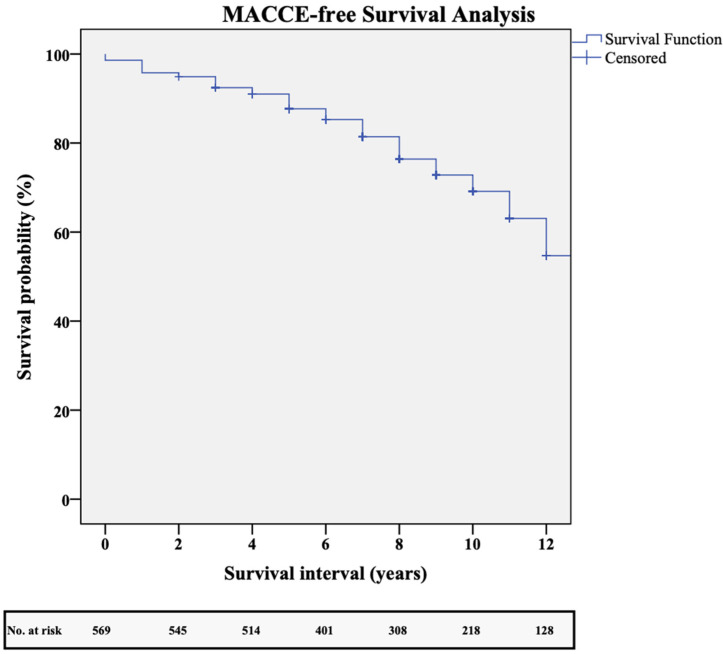
Kaplan–Meier major adverse cardiac and cerebrovascular event (MACCE)-free survival analysis.

**Table 1 jcm-14-01707-t001:** Baseline patient characteristics.

Variable	No (%)/Median (IQR, Q1–Q3)
Age (y)	69 (61–76)
Male gender	550 (92.1)
BMI	26.3 (24.3–28.4)
EuroSCORE II	1.5 (1–2.2)
COPD	37 (6.2)
Tobacco use	124 (20.8)
AF	80 (13.5)
PVD	54 (9)
Medically treated type II diabetes	113 (18.9)
CKD (GFR < 50 mL/min/1.73 m^2^)	74 (12.4)
Renal replacement therapy	5 (0.8)
Carotid stenosis	67 (11.2)
Previous CVA	26 (4.4)
Previous MI	
<48 h	3 (0.5)
48 h−21 d	65 (10.9)
21 d−91 d	49 (8.2)
>91 d	68 (11.4)
Previous PCI	167 (28)
Coronary angiography data	
1 VD	56 (9.4)
2 VD	318 (53.3)
3 VD	223 (37.3)
LVEF (%)	
>50	488 (81.7)
30–50	97 (16.3)
<30	12 (2)
Priority	
elective	322 (53.9)
urgent	275 (46.1)

Values are presented as no (%) or median and IQR (interquartile range) Q1–Q3, Q-quartile; AF, atrial fibrillation; BMI, body mass index; CKD, chronic kidney disease; CVA, cerebrovascular accident; COPD, chronic pulmonary disease; EuroSCORE, European System for Cardiac Operative Risk Evaluation; GFR, glomerular filtration rate; LVEF, left ventricular ejection fraction; MI, myocardial infarction; PCI, percutaneous coronary intervention; PVD, peripheral vascular disease; VD, vessel disease.

**Table 2 jcm-14-01707-t002:** Operative data.

Variable	No (%)/Median (IQR, Q1–Q3)
Number of grafts	
2	523 (87.6)
3	74 (12.4)
Use of LITA	597 (100)
Use of RITA	33 (5.5)
Use of RA	411 (68.8)
Use of GSV	47 (7.9)
Aortic touch	2 (0.3)
Number of total arterial revascularisations	552 (92.5)
Conversion to sternotomy	8 (1.3)
Ventilation time (hours)	2 (0–5)
RBC transfusion (per patient)	0 (0–0)
Number of patients with RBC transfusion	63 (10.6)
FFP transfusion (per patient)	0 (0–0)
Number of patients with FFP transfusion	12 (2)
Platelet transfusion (per patient)	0 (0–0)
Number of patients with platelet transfusion	15 (2.5)
Number of grafts to the anterior wall	
0	5 (0.8)
1	407 (68.2)
2	185 (31)
Number of grafts to the posterior wall	
0	553 (89.3)
1	64 (10.7)
Number of grafts to the lateral wall	
0	210 (35.2)
1	374 (62.6)
2	13 (2.2)
Complete revascularisation	539 (90.3)
Planned as hybrid procedures	61 (10.2)

Values are presented as no (%) or median and IQR (interquartile range) Q1–Q3, Q-quartile; FFP, fresh-frozen plasma; GSV, great saphenous vein; LITA, left internal thoracic artery; RA, radial artery; RBC, red blood cell; RITA, right internal thoracic artery.

**Table 3 jcm-14-01707-t003:** Perioperative results.

Variable	No (%)/Median (IQR, Q1–Q3)
New renal failure requiring dialysis	6 (1.0)
New-onset AF	13 (2.2)
Postoperative CVA	2 (0.3)
SSI	10 (1.7)
Postoperative CPR	5 (0.8)
Postoperative MI	7 (1.2)
Postoperative PCI as a revision	5 (0.8)
Postoperative PCI planned as part of hybrid procedure (30 d)	8 (1.3)
Reoperation for bleeding	13 (2.2)
Reoperation with bypass revision	5 (0.8)
Postoperative ventricular arrhythmia	6 (1.0)
Length of ICU stay (d)	1 (1–1)
Length of hospital stay (d)	7 (6–8)
30 d mortality	3 (0.5)

Values are presented as no (%) or median and IQR (interquartile range) Q1–Q3, Q-quartile; AF, atrial fibrillation; CPR, cardiopulmonary resuscitation; CVA, cerebrovascular accident; ICU, intensive care unit; MI, myocardial infarction; PCI, percutaneous coronary intervention; SSI, surgical site infection.

**Table 4 jcm-14-01707-t004:** Cox regression analysis for late mortality and MACCEs.

Variable	Late Mortality			MACCE		
	*p* Value	HR	95% CI	*p* Value	HR	95% CI
LVEF < 30%	0.042	2.652	1.037–6.780	0.005	3.441	1.444–8.198
LVEF 30–50%	0.025	1.753	1.072–2.866	0.018	1.556	1.078–2.246
PVD	0.002	2.306	1.365–3.894	0.010	1.736	1.142–2.461
CKD	0.001	2.461	1.448–4.182	<0.001	2.271	1.520–3.393
Previous PCI	0.002	0.390	0.215–0.709			
Carotid stenosis	0.029	1.783	1.060–3.001			
CVA	<0.001	3.581	1.964–6.529	0.003	2.298	1.333–3.961

CI, confidence intervals; CKD, chronic kidney disease; CVA, cerebrovascular accident; HR, hazard ratio; LVEF, left ventricular ejection fraction; MACCE, major adverse cardiac and cerebrovascular event; PCI, percutaneous coronary intervention; PVD, peripheral vascular disease.

## Data Availability

The raw data supporting the conclusions of this article will be made available by the authors on request.

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
