# Peer review of "Medium- and Long-Term Outcomes of 597 Patients Following Minimally Invasive Multi-Vessel Coronary Off-Pump Bypass Surgery"

_jcm, 2025, doi:10.3390/jcm14051707_

Round 1

Reviewer 1 Report

Comments and Suggestions for Authors

The authors present their single-center retrospective series of almost 600 patients undergone off-pump MICS CABG over a time span of 12 years.

Mean follow-up is 8 years and > 95% complete.

The study is well written and the results presented are very good in terms of complications, mid/long-term mortality and MACCE rates.

Noteworthy is the extremely high rate of total arterial revascularization, which, in my opinion, is the leading way in contemporary coronary surgery and hopefully its superiority will be confirmed soon by the early results of the ROMA trial.

Other important features include, a fast-track extubation protocol in the OR, selective nerve blockade and endoscopic graft harvesting in order to reduce the global burden associated with any cardiac operation.

Finally, I strongly believe, as the Authors, in the systematic use of bypass flow measurement which is an indispensable quality control tool and also plays an important role in case of medico-legal issues.

Apart from its retrospective single-center design, a limit of the study is that provides only clinical outcomes without investigating long-term grafts permeability.

The authors declare that the study has been designed to include all consecutive patients, however it seems that a certain selection has been made, apart from the declared exclusion criteria.

 This is consistent with the low Euroscore II as well as the low percentage of patients with diabetes, COPD or chronic renal disease.

Also, the vast majority of patients have a >50% (normal) LVEF: 81.7%.

In other words, my impression is that the cohort of the study is way better than a real-word, all-comers CABG population.

Perhaps, apart from the learning curve, the uncertainty of obtaining such good results in worse CABG patients (the ones who theoretically would benefit more of a minimally invasive approach) is the main reason limiting the spread of these techniques around the world.

Could you please provide a comment on that?

The Authors report a 90.3% of complete revascularization and stated (ref 12) that an incomplete revascularization leads to reduced long-term survival (not significant after risk adjustement). Could you explain the reasons of incomplete revascularization cases (around 10%) and how they were treated?

In your population 92.1% of patients were male. How can you explain this figure, higher than other reports (males around 72-80% of CABG, according to literature)?

Line 17, page 1: I would change “bypass transplant”, it sounds weird.

The authors stated that they switched to robotic surgery at some stage.

The number of cases performed with this approach is not reported.

Also, it would be interesting to compare, not necessarily in this paper,  the results of the two techniques since they differ significantly (at least reporting the main outcomes).

Author Response

The authors present their single-center retrospective series of almost 600 patients undergone off-pump MICS CABG over a time span of 12 years.

Mean follow-up is 8 years and > 95% complete.

The study is well written and the results presented are very good in terms of complications, mid/long-term mortality and MACCE rates.

Noteworthy is the extremely high rate of total arterial revascularization, which, in my opinion, is the leading way in contemporary coronary surgery and hopefully its superiority will be confirmed soon by the early results of the ROMA trial. 

Other important features include, a fast-track extubation protocol in the OR, selective nerve blockade and endoscopic graft harvesting in order to reduce the global burden associated with any cardiac operation.

Finally, I strongly believe, as the Authors, in the systematic use of bypass flow measurement which is an indispensable quality control tool and also plays an important role in case of medico-legal issues.

1. Apart from its retrospective single-center design, a limit of the study is that provides only clinical outcomes without investigating long-term grafts permeability.

1. Thank you for your accurate observation; we exclusively offer clinical outcomes, as examining the long-term permeability of grafts would require an ethical vote and patient approval for either coronary computed tomography or coronary angiography. Both investigative methods expose the patient to radiation and contrast agents, along with potential complications associated with angiography, including peripheral access vessel issues, coronary complications, aortic dissection, and allergic reactions to the contrast agent.

We had mentioned already this as limitation Lines 271-272.

2. The authors declare that the study has been designed to include all consecutive patients, however it seems that a certain selection has been made, apart from the declared exclusion criteria.

 This is consistent with the low Euroscore II as well as the low percentage of patients with diabetes, COPD or chronic renal disease.

Also, the vast majority of patients have a >50% (normal) LVEF: 81.7%.

In other words, my impression is that the cohort of the study is way better than a real-word, all-comers CABG population. 

Perhaps, apart from the learning curve, the uncertainty of obtaining such good results in worse CABG patients (the ones who theoretically would benefit more of a minimally invasive approach) is the main reason limiting the spread of these techniques around the world.

Could you please provide a comment on that?

2. Thank you once more for this remark. Indeed, the patients appear to be healthier than a general coronary population; nonetheless, the primary criterion for selecting patients for this method is the patient's coronary anatomy and the localization of the stenoses. We do not exclude patients with a greater burden of comorbidities or diminished left ventricular ejection fraction. We concur with your assessment that these individuals are indeed the primary beneficiaries of this strategy. We added an explanation Lines 42-45.

3. The Authors report a 90.3% of complete revascularization and stated (ref 12) that an incomplete revascularization leads to reduced long-term survival (not significant after risk adjustement). Could you explain the reasons of incomplete revascularization cases (around 10%) and how they were treated?

3. I appreciate this excellent inquiry. Slightly over ten percent of the procedures were hybrid planned; however, as the study extends over 13 years, commencing in 2008, it remains uncertain whether patients who reported receiving a PCI procedure followed by stent implantation during follow-up were undergoing the originally intended procedure as part of the hybrid treatment or if it resulted from a failed prior stent, unsuccessful bypass, or de novo stenosis. Consequently, we opted to classify the case as incomplete and to categorise the procedure during follow-up as renewed revascularisation, also noting it as MACCE positive. We have gained significant insights from analysing these data, and we are now thorough in organising and conducting the PCI step in a hybrid case.

We added this comment as a new paragraph under ‘Discussion’, Lines 309-315.

4. In your population 92.1% of patients were male. How can you explain this figure, higher than other reports (males around 72-80% of CABG, according to literature)?

4. Thank you for this question. The only explanation for the reduced number of included female patients is that they often present with diffuse and small coronaries which we mentioned as exclusion criterion (Line 57).

5. Line 17, page 1: I would change “bypass transplant”, it sounds weird.

5. Thank you for this observation, we changed the wording to ‘bypass graft’ (Line 17)

6. The authors stated that they switched to robotic surgery at some stage.

The number of cases performed with this approach is not reported.

6. Thank you for this great question. In the present study in 31 cases were the grafts robotic assisted harvested. We added this information in the ‘Results’ chapter, Line 167-168.

7. Also, it would be interesting to compare, not necessarily in this paper,  the results of the two techniques since they differ significantly (at least reporting the main outcomes).

7. Thank you for this great observation. We are currently preparing a paper on this topic.

Reviewer 2 Report

Comments and Suggestions for Authors

Dear Authors,

I wish to congratulate you on the results of your study presented in the manuscript: “Medium and long-term outcomes of 597 patients following minimally invasive multi-vessel coronary off-pump bypass surgery”

The manuscript is well-written, and the presented results are impressive.

I would suggest adding some Figures presenting the surgical technique to help potential readers understand the practical aspect of the surgical techniques you’ve described.

I would add study limitation

Kind regards

R

Author Response

I wish to congratulate you on the results of your study presented in the manuscript: “Medium and long-term outcomes of 597 patients following minimally invasive multi-vessel coronary off-pump bypass surgery”

The manuscript is well-written, and the presented results are impressive.

1.I would suggest adding some Figures presenting the surgical technique to help potential readers understand the practical aspect of the surgical techniques you’ve described.

1.This is an excellent suggestion. We added 5 Pictures to the description of the surgical technique.

2.I would add study limitation.

2.This is also a great observation. We had added a limitations chapter to the discussion section, only not added a subtitle to it, lines 265-275.